

# How much genetic variation is stored in the endangered and fragmented shrub *Tetraena mongolica* Maxim?

Yingbiao Zhi[1,2,3,*], Zhonglou Sun[2,*], Ping Sun[2], Kai Zhao[4], Yangnan Guo[5,6], Dejian Zhang[5] and Baowei Zhang[2]

[1] School of Ecology and Environment, Inner Mongolia University, Hohhot, Inner Mongolia, China
[2] School of Life Sciences, Anhui University, Hefei, Anhui, China
[3] Ordos Institute of Technology, Ordos, Inner Mongolia, China
[4] School of Resource and Environment, Anqing Normal University, Anqing, Anhui, China
[5] School of Life Sciences, Inner Mongolia University, Hohhot, Inner Mongolia, China
[6] China Energy Technology Holdings Ltd., Beijing, China
[*] These authors contributed equally to this work.

## ABSTRACT

*Tetraena mongolica* Maxim (Zygophyllaceae) is an endangered species endemic to western Inner Mongolia and China, and is currently threatened by habitat loss and human over-exploitation. We explored the genetic background, its genetic diversity, population structure, and demographic history, based on 12 polymorphic nuclear microsatellite loci. Our results indicated high genetic diversity in extant populations, but no distinguishable gene cluster corresponding with a specific biogeography. Population demography analysis using a MSVAR indicated a strong, recent population decline approximately 5,455 years ago. These results suggest that the Yellow River and Zhuozi Mountain range may not prevent pollination between populations. Finally, we surmised that the population demography of *T. mongolica* was likely to have been affected by early mankind activities.

## INTRODUCTION

Understanding population history and genetic structure is a key aspect of ecological research (*Rockwood, 2006*). Endemic species with restricted geographic distributions have become a central concern of biologists faced with the problem of preserving rare species endangered by habitat destruction and fragmentation (*Ge et al., 2003*). For endemics with narrow ranges and declining populations, information about historical patterns of demography, genetic structure, and genetic variation within and among natural populations helps to clarify population structure, and the organism's evolutionary history, supporting conservation and management efforts (*Moritz, 1994*; *Ge et al., 2011*). Intraspecific genetic variation is the most fundamental level of biodiversity, providing the basis for evolutionary change and the ability of species to adapt to new environmental conditions (*Frankham, Ballou & Briscoe, 2002*). In contrast, plants and invertebrates with high fecundity and human-mediated

Corresponding authors
Yingbiao Zhi, zyb.china@163.com
Baowei Zhang, zhangbw@ahu.edu.cn

dispersal ability, its populations can be successfully reestablished and will experience range expansion (*Song et al., 2013*).

Previous studies revealed that natural landscape features such as mountains and rivers can act as genetic boundaries and shape the structure of populations (*Funk et al., 2001*; *Whiteley, Spruell & Allendorf, 2004*). However, anthropogenic landscape features also have an impact on genetic structure (*Gaines et al., 1997*) and population dynamics (*Nupp & Swhart, 1998*). Anthropogenic disturbance, such as roads, have dramatically increased the physical isolation of populations, and it has been assumed that such isolation will lead to reduced gene flow and consequently reduced genetic diversity in populations (*Byrne et al., 2007*). Furthermore, habitat destruction and land-use change may also influence gene flow at the landscape scale (*Manel et al., 2003*; *Eckstein et al., 2006*). These anthropogenic effects may also occur in the center of a species' range and may thus be superimposed on natural geographic patterns.

*Tetraena mongolica* Maxim is a member of the broader genus *Tetraena* in the subfamily Zygophyllaceae (*Beier, Chase & Thulin, 2003*; *Lauterbach et al., 2016*), and is endemic to western Inner Mongolia around the Yellow River basin, and is nationally endangered in China (*Fu, 1992*; *Xu et al., 1998*; *Zhang & Yang, 2000*). Its distribution is restricted to the western Gobi, the largest desert in Asia, and one characterized by extremely low annual rainfall (*Xu et al., 1998*; *Zhang & Yang, 2000*), where *T. mongolica* is able to survive because of its extensive root system, and acts as a windbreak and soil stabilizer (*Dong & Zhang, 2001*; *Zhang et al., 2003*). Its stems contain high levels of waxes and oils (*Wang, Ma & Zheng, 2000*), and are combustible, even when green. For this reason, *T. mongolica* is a popular firewood species, and its range has declined alarmingly through overexploitation (*Zhang & Yang, 2000*; *Ge et al., 2011*). Based on inter-simple sequence repeat (ISSR) markers, *Ge et al. (2003)* revealed that this species presents an intermediate level of intraspecific genetic diversity despite its limited distribution. Moreover, *Ge et al. (2003)* discovered that there was low genetic differentiation among *T. Mongolic* populations, which was due to the extensive gene flow within this population. However, neither the impacts of natural barriers to dispersal, nor human influences on the genetic structure and demographic history of *T. mongolica* have been ascertained.

Evolutionary, demographic and genetic analyses all contribute to conservation and management of species (*Beaumont & Bruford, 1999*; *O'Brien, 1994*). We generated a comprehensive genetic characterization for *T. mongolica* with the aim of supporting a conservation strategy. We used twelve microsatellites SSRs (Simple Sequence Repeats) genotyped onto an extensive dataset to evaluate the current genetic diversity in *T. mongolica* populations, and to assess the effect of natural landscape barriers (Yellow River and Zhuozi Mountains) in shaping population structure. Lastly, we modeled the demographic history of *T. mongolica* to assess the effects of historic events on population demography. Our findings may be useful for the conservation and management of *T. mongolica* and other species endemic to the Yellow River basin.

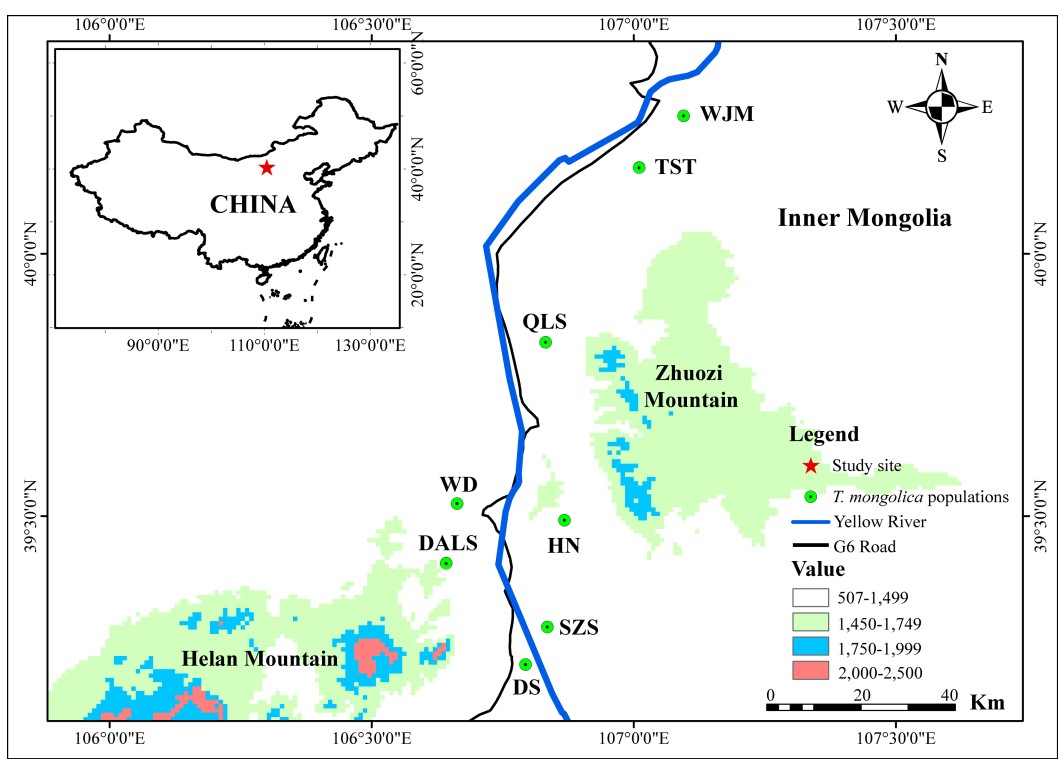

**Figure 1** Map showing the population location of *T. mongolica* sampled in this study.

# MATERIAL AND METHODS

## Ethical statement

The collection of samples was performed within an investigation project on plants of *T. mongolica*. This investigation project and the sample collection were approved by the West Ordos National Nature Reserve, Inner Mongolia Province, China. Field experiments were also approved by the West Ordos National Nature Reserve, Inner Mongolia Province, China.

## Sample collection

Between 2010 and 2014, 339 leaf samples of *T. mongolica* were collected from eight populations along the G6 Road: Shizuishan (SZS, $N = 32$); Dishan (DS, $N = 64$); Hainan (HN, $N = 51$); Dongalashan (DALS, $N = 62$); Wuda (WD, $N = 32$); Qianlishan (QLS, $N = 32$); Wujiamiao (WJM, $N = 36$) and Taositu (TST, $N = 30$) (Fig. 1). Leaves sample were powdered in liquid nitrogen and stored frozen at $-80\ °C$.

## DNA extraction, PCR amplification and microsatellite genotyping

Total genomic DNA was extracted from the powdered tissue following a modified CTAB procedure (*Doyle & Doyle, 1987*), and purified via an EasyPure PCR Purification Kit (TransGene). In the present study, we used twelve high polymorphic loci for *T. mongolica* (*Zhi et al., 2014*) as genetic markers. PCR reaction mixtures (25 μL) consisted of 1 μL
genomic DNA (concentration 10–50 ng/µL), 2 µL 10× buffer, 1 µL of 2.5 mM MgSO$_4$, 2 µL of 2 mM dNTPs, 1 U *Taq* polymerase, 0.3 mM of each primer (forward primer fluorescently labeled with FAM, HEX or TAMRA) and sufficient water. The amplification program was conducted with following conditions: 5 min denaturing at 95 °C; followed by 35 cycles of 30 s at 95 °C, 20 s at the annealing temperature (55/60 °C), 30 s at 72 °C; and 5 min at 72 °C. PCR products were genotyped on an ABI 3730 semi-automated sequencer (PE Applied Biosystems, Foster City, CA, USA) utilizing the GS500 marker, followed by analysis under GeneMarker 1.85 (SoftGenetics LLC, State College, PA, USA) (*Holland & Parson, 2011*).

## Data analysis

The presence of null alleles and genotyping errors in microsatellite genotyping was detected by Micro-Checker v2.2.3 as previously described by *Van Oosterhout et al. (2004)*, while the linkage disequilibrium was tested with GENEPOP 4.2.1 as described by *Rousset (2008)*. In addition, several population genetic summary statistics to describe genetic variation were estimated by GENETIX v.4.02 as described in *Belkhir, Borsa & Chikhi (2001)*, including mean number of alleles per locus (MNA), observed heterozygosities ($H_O$), expected heterozygosities ($H_E$) and inbreeding coefficients ($F_{IS}$). In addition, allelic richness (AR) was also calculated to estimate the allelic diversity that compensates for unequal sample size by FSTAT and averaged across loci (*Goudet, 2002*). Genetic differentiation ($F_{ST}$) between populations was estimated using ARLEQUIN 3.0 (*Excoffier, Laval & Schneider, 2005*), and statistical significance of $F_{ST}$ values was tested with 10,000 permutations. In addition, the association between the estimates of $F_{ST}$/ 1–$F_{ST}$ (*Rousset & Raymond, 1997*) and land-based Manhattan distance were assessed using the Mantel test, implemented in the Isolation by Distance Web Service (IBDWS) software (*Jensen, Bohonak & Kelley, 2005*); the statistical significance of the values was obtained by 10,000 randomization steps.

A Bayesian analysis of population structure as previously described in *Pritchard, Stephens & Donnelly (2000)* was carried out to estimate the number of potential clusters present in the microsatellite data, and to assign individuals to inferred clusters by STRUCTURE. Specifically, five independent runs were carried for different values of *K* between 1 and 8, using no prior information about individual location, and assuming admixture and correlated allele frequencies. The Markov Chain Monte Carlo (MCMC) was run for a total of 1 million generations discarding the first 100,000 as burn-in. The most likely *K* explaining the variation in the data was selected estimating the maximal value of the log likelihood [Ln Pr(X/K)] of the posterior probability of the data for a given *K* (*Pritchard, Stephens & Donnelly, 2000*), and the Δ*K* statistic (*Evanno, Regnaut & Goudet, 2005*), as implemented in the program Structure Harvester version 0.6.94 (*Earl & VonHoldt, 2012*). The population structure results were graphically displayed by the software DISTRUCT (*Rosenberg, 2004*). In addition, we visualized the genetic differentiation among all samples with a factorial correspondence analysis (FCA) in GENETIX version 4.0. Furthermore, we constructed a population graph network described by *Dyer & Nason (2004)* using the popgraph package (*Dyer, 2014*) in R 2.15.3 (*R Development Core Team, 2013*). The method is based on the genetic covariance structure among populations analyzed simultaneously

(*Dyer & Nason, 2004*). Populations that exhibit significant genetic matrix correlation will be connected in the network by edges (lines), and the length of the edges is inversely proportional to the genetic covariance between the populations. Therefore, longer edges indicate lower genetic covariance between populations. Populations that are not connected indicate the absence of migration, and the presence of subgraphs (a smaller network within a large network) indicates that a population or group of populations maintain a weak or null genetic connection (*Dyer, 2007*; *Dyer & Nason, 2004*; *Dyer, Nason & Garrick, 2010*).

Demographic history was performed in BOTTLENECK 1.2.02 (*Piry, Luikart & Cornuet, 1999*) and assessed using Wilcoxon's sign rank test and mode-shift test as previously described in *Cornuet & Luikart (1996)* and *Luikart & Cornuet (1998)*, respectively. The software MSVAR v.1.3 was used to characterize the recent demographic history of the whole *T. mongolica* population based on the microsatellite data as described in *Storz, Beaumont & Alberts (2002)*. Specifically, this method assumes that a current population (of size $N_0$) passed through a demographic change (a bottleneck or an expansion) at time $T$ in the past, which changed its size from $N_1$ to $N_0$ following an exponential model. Five independent simulations were run to estimate the distributions of these three parameters. For *T. mongolica*, the average generation time is four years (*Xu et al., 1998*), and this period was adopted for the simulation. Each MSVAR run consisted of $2 \times 10^9$ iterations of the MCMC algorithm discarding the first 10% of the coalescent simulations as burn-in. The median (50%) of the posterior distributions were calculated from five runs data. Finally, we plotted the marginal posterior distributions of the three parameters by the LOCFIT package (*Loader, 2007*) implemented in R based on five runs.

## RESULTS

### Genetic diversity

In this study, a total of 339 individuals were genotyped at 12 loci. Micro-Checker did not indicate null alleles or genotyping errors such as large allele dropout or stuttering. There was no linkage disequilibrium at any locus in any population. The MNA for the eight populations varied between 13.17 and 17.67 with an overall value of 15.25. The overall observed heterozygosity ($H_O$) was 0.840 (0.810–0.873), while the overall expected heterozygosity ($H_E$) was 0.868 (0.832–0.882) (Table 1). Allelic richness (AR) ranged from 6.860–10.529, with an overall allelic richness across loci of 9.382 (Table 1). Inbreeding coefficient analysis generated negative values in SZS and QLS populations (Table 1). For the population as a whole, genetic diversity as characterized by microsatellite markers was higher than that reported for other shrub species (Table 2).

### Population structure and genetic relationship

Based on STRUCTURE analysis, the Dealt $K$ statistics output showed a clear maximum at $K = 2$ (Delta $K = 4.98$) (Fig. 2B), but no obvious maximum log likelihood of posterior probability was found (Ln$P(K) = -19,881.82$) (Fig. 2A). The $\Delta K$ value was remarkable at $K = 6$ (Delta $K = 3.69$) (Fig. 2B), with an obvious maximum log likelihood of posterior probability (Ln$P(K) = -19,596.44$) (Fig. 2A). These data suggest that six potential genetic clusters may exist among them. Notably, factors such as recent admixture, admixture

**Table 1  Genetic variability observed within populations using nuclear microsatellite loci.**

| Population | $N$ | MNA | AR | $H_O$ | $H_E$ | $F_{IS}$ (IC 95%) |
|---|---|---|---|---|---|---|
| SZS | 32 | 13.17 | 6.860 | 0.850 | 0.832 | −0.00163 (−0.03378–0.00409) |
| DS | 64 | 17.67 | 8.284 | 0.836 | 0.881 | 0.06051 (0.02421–0.07440) |
| NH | 51 | 17.67 | 10.499 | 0.812 | 0.880 | 0.08889 (0.04226–0.09998) |
| DALS | 62 | 16.75 | 8.974 | 0.810 | 0.867 | 0.07559 (0.04021–0.09279) |
| WD | 32 | 14.67 | 8.829 | 0.851 | 0.867 | 0.03657 (−0.02664–0.07773) |
| QLS | 32 | 13.92 | 8.532 | 0.873 | 0.854 | −0.00596 (−0.05695–0.00596) |
| WJM | 36 | 15.67 | 10.529 | 0.852 | 0.882 | 0.04891 (−0.01434–0.05576) |
| TST | 30 | 14.08 | 9.397 | 0.840 | 0.877 | 0.06018 (0.02117–0.06018) |
| Total | 339 | 15.45 | 9.382 | 0.840 | 0.868 | 0.06760 (0.05360–0.06888) |

**Notes.**

$N$, number of individuals; MNA, mean number of allele per locus; AR, allelic richness; $H_o$ and $H_E$, observed and expected heterozygosity; $F_{IS}$, inbreeding coefficient.

**Table 2  Genetic diversity of *Tetraena mongolica* and other shrub based on nuclear microsatellite loci.**

| Species | $N$ | MNA | $H_O$ | $H_E$ | Reference |
|---|---|---|---|---|---|
| *Tetraena mongolica* | 339 | 15.45 | 0.84 | 0.868 | In this study |
| *T. mongolica* | 338 | 1.6 | 0.199 | 0.345 | *Zhang & Yang (2000)* |
| *Zygophyllum xanthoxylon* | 61 | 2.2 | 0.43 | 0.392 | *Zhang & Yang (2000)* |
| *Ziziphus celata* | 595 | 2.23 | 0.69 | 0.39 | *Gitzendanner et al. (2012)* |
| *Adiantum capillus-veneris* | 151 | – | 0.13–0.37 | 0.2–0.63 | *Pryor et al. (2001)* |
| *Grevillea macleayana* | 321 | – | 0.248-0.523 | 0.420–0.523 | *England et al. (2002)* |
| *Arabidopsis lyrata* | 344 | 9.3 | 0.48 | 0.52 | *Clauss & Mitchell-Olds (2006)* |
| *Calothamnus quadrifidus* | 114 | 19.67 | 0.584 | 0.867 | *Byrne et al. (2007)* |
| *Myrtus communis* | 48 | – | 0.258–0.802 | 0.125–0.875 | *Albaladejo et al. (2010)* |
| *Schiedea adamantis* | 49 | – | 0.125–0.755 | 0.041–0.787 | *Culley et al. (2008)* |

with unsampled/unobservable "ghost" populations, and recent bottlenecks may lead to misinterpretation of STRUCTURE results (*Gilbert et al., 2012*; *Lawson et al., 2012*; *Falush, Van Dorp & Lawson, 2016*). According to this framework, $K = 6$ may be a pseudophase. The highly mixed color bars in the DISTRUCT diagram (for $K = 2$–6, Fig. 2C) indicated strong admixture among the eight populations. Furthermore, $F_{ST}$ values among these populations ranged from 0.00034 to 0.04284, indicating a weak genetic differentiation across them (Table 3). Besides, IBD tests detected no significant correlation between geographical distances and genetic distance for the whole sampling ($r = 0.0608, p \leq 0.3940$). Furthermore, no separate groups were identified in the FCA analysis (Fig. 3). Specifically, all populations were highly clumped and overlapped (Fig. 3). The popgraph software produced a population network with no subgraphs (Fig. 4). Overall, the population network exhibited high genetic connection among the cohorts, where each population was connected to at least four other populations.

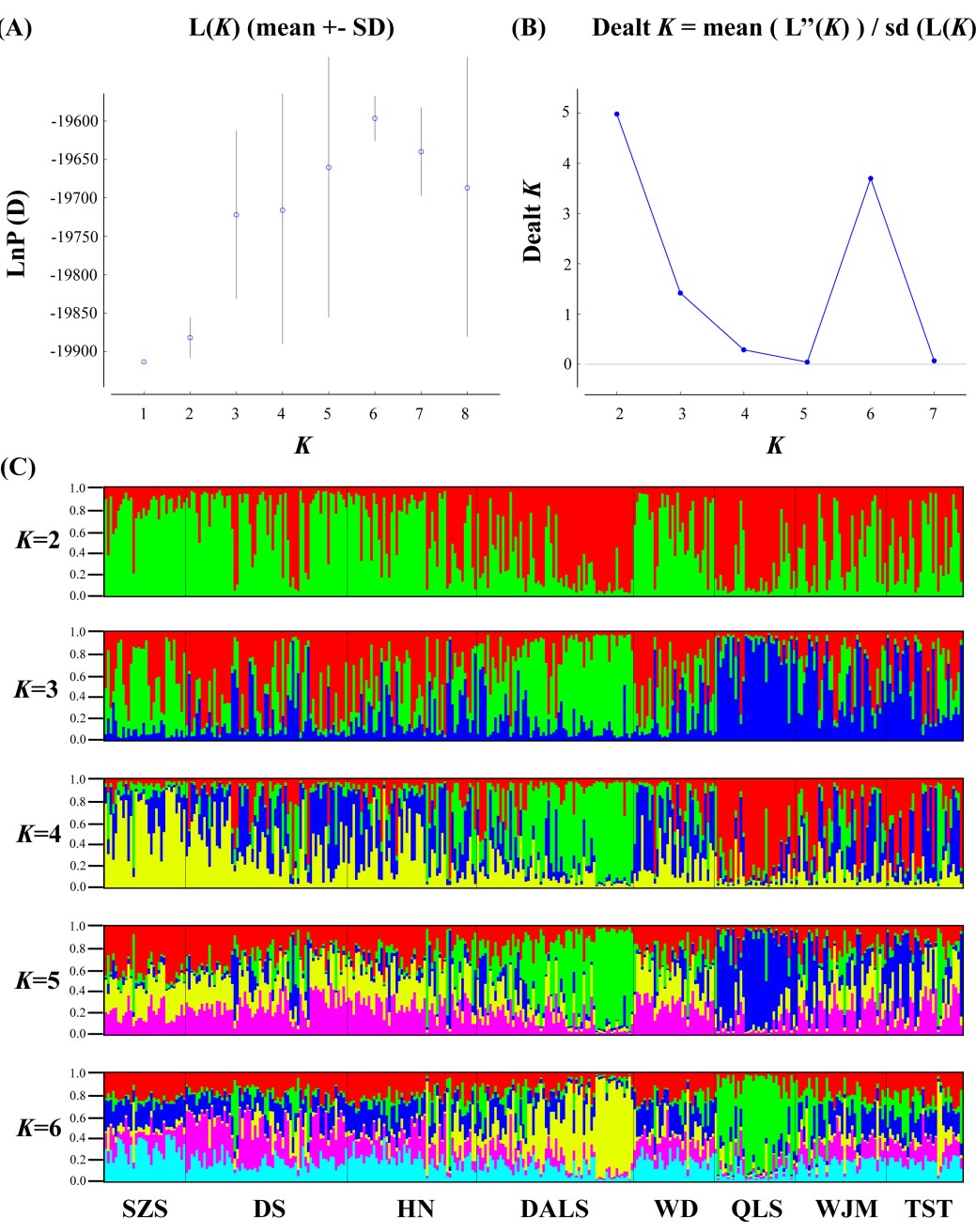

**Figure 2  Bayesian STRUCTURE clustering results of microsatellite variation among *T. mongolica* populations.**  (A) The linear relationship between LnP(D) and *K*, (B) Dealt *K* values as a function of *K* based on five runs and (C) STRUCTURE output from *K* = 2 to 6.

## Population demography

In present study, there is no significant signal of recent bottlenecks in eight populations under both TPM and SMM model whilst the mode-shift test also showed a normal L-shaped distribution of allele frequencies. However, the MSVAR results showed the posterior distribution of $N_0$ and $N_1$ did not overlap under exponential models, which indicates that

**Table 3  Pairwise $F_{ST}$ estimates based on nuclear microsatellite loci.**

| Populations | 1 | 2 | 3 | 4 | 5 | 6 | 7 | 8 |
|---|---|---|---|---|---|---|---|---|
| 1. SZS | | | | | | | | |
| 2. DS | 0.01599* | | | | | | | |
| 3. HN | 0.00607* | 0.00620* | | | | | | |
| 4. DALS | 0.01648* | 0.01623* | 0.0076* | | | | | |
| 5. WD | 0.01795* | 0.00834* | 0.00793* | 0.01522* | | | | |
| 6. QLS | 0.04284* | 0.02743* | 0.01740* | 0.02972* | 0.03673* | | | |
| 7. WJIM | 0.01580* | 0.00944* | 0.00034 | 0.00769* | 0.01095* | 0.01839* | | |
| 8. TST | 0.02699* | 0.01837* | 0.01373* | 0.01829* | 0.02349* | 0.02138* | 0.00912* | |

Notes.
*mean $P < 0.05$.

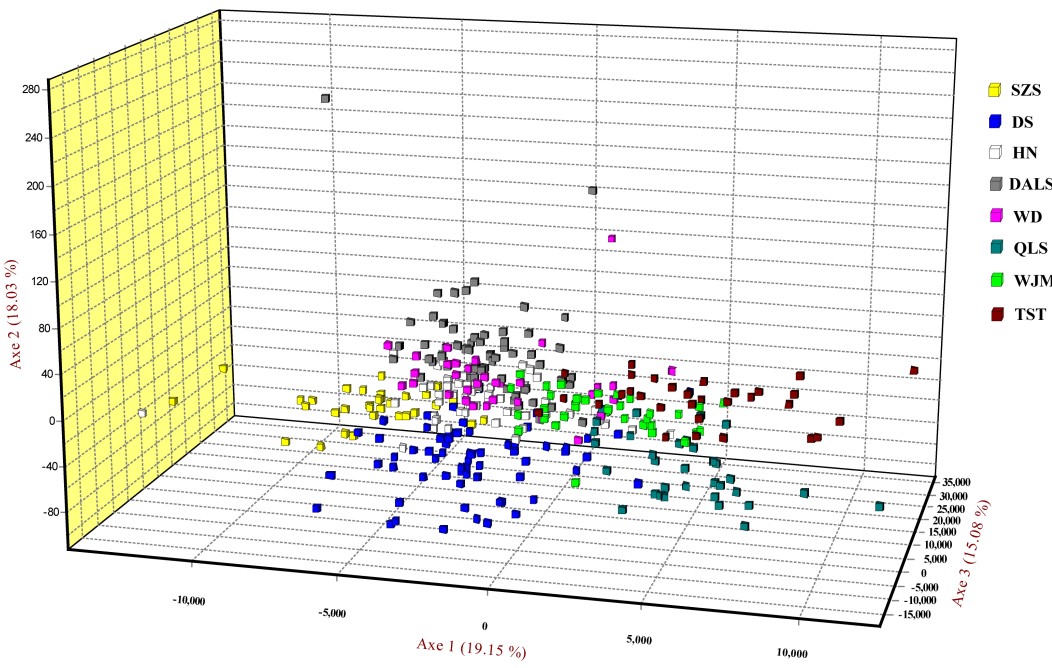

**Figure 3  Factorial correspondence analysis performed for *T. mongolica* based on nuclear microsatellite loci.** Symbols and colors represent individuals from different populations.

the whole population passed through a significant reduction in effective population size (Fig. 5A). Statistically, the average medians of the posterior distributions were approximately 2.9652 for $\log N_0$, and approximately 4.7938 for $\log N_1$ (Fig. 5A). Therefore, for the present *T. mongolica* population, the current effective population size ($N_0$) was approximately 923, while the ancestral effective population size ($N_1$) was approximately 62,214, showing an approximately 67-fold population decrease. Furthermore, the medians of the posterior distribution $\log T = 3.7368$ (Fig. 5B), indicate a recent population decline took place approximately 5,455 years ago.
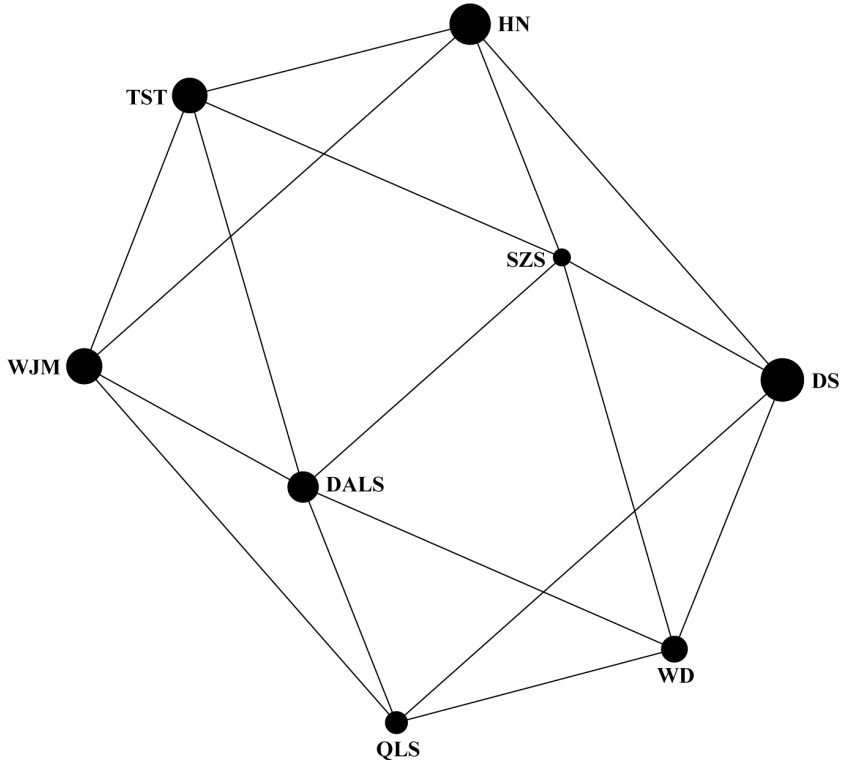

**Figure 4** **Population graph for 8 populations of *Tetraena mongolica* based on nuclear microsatellites data.** The size of the nodes (spheres) represents the genetic variation within populations and edges (lines) connect directly two populations showing significant genetic covariance.

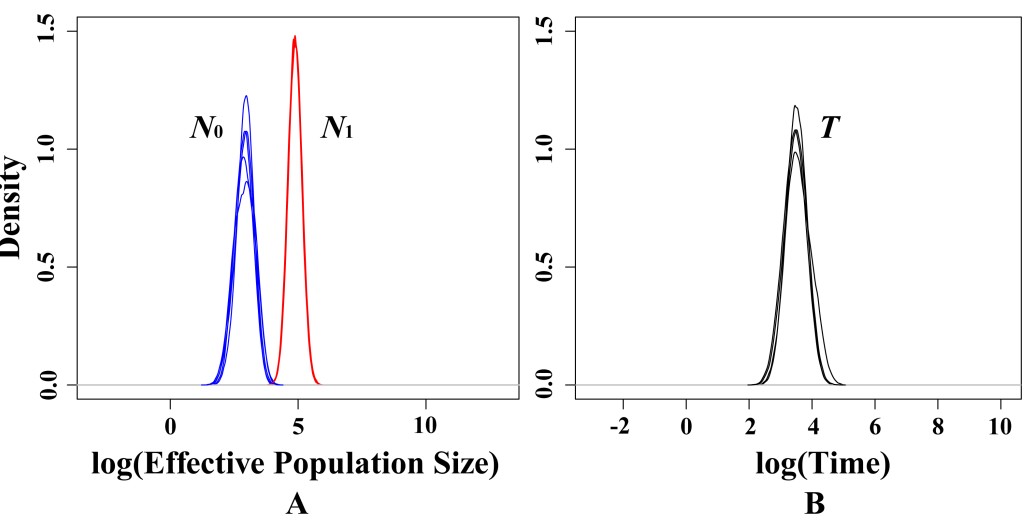

**Figure 5** **Estimated posterior distributions of $N_0$, $N_1$ and $T$ using MSVAR.** $N_0$, current effective population sizes (blue curve); $N_1$, ancestral effective population sizes (red curve); $T$, time since population change (black curve). All densities are represented in a log10 scale.

## DISCUSSION

### Genetic diversity

Because of human overexploitation, *T. mongolica* has undergone a dramatic population decline in past decades (*Ge et al., 2011*). However, our assessment of genetic variation based on microsatellite data reveals high levels of genetic diversity in this species. In the population as a whole, high microsatellite diversity was detected, with MNA, $H_O$ and $H_E$ values of 15.45, 0.84 and 0.868, respectively (Table 1). Based on inter-simple sequence repeats (ISSR) marker, this species' average gene diversity was estimated to be 0.177 within populations ($H_E$), and the $H_O$ ranged from 0.213 to 0.305, with an average of 0.263 at the population level (*Ge et al., 2003*). Compared with *Ge et al. (2003)* study, we determined there to be extremely high genetic diversity based on nuclear microsatellites in this species. Therefore, SSR may represent a more advantageous alternative to assess genetic diversity than ISSR in *T. mongolica*. However, SSR may also be over-estimating genetic diversity as we detected higher genetic diversity in *T. mongolica* compared with other shrub species such as *Zygophyllum xanthoxylon*, *Ziziphus celata*, *Adiantum capillus-veneris*, *Grevillea macleayana*, *Arabidopsis lyrata*, *Calothamnus quadrifidus*, *Myrtus communis* and *Schiedea adamantis* (Table 2). Generally, species with high genetic diversity are members of large populations that were geographically widespread in recent history (*González et al., 1998*). In this study, the high genetic diversity of *T. mongolica* may reveal the large effective size of ancestral populations, as supported by the demographic analysis using MSVAR. Furthermore, from a conservation perspective, it also implies that the recent sharp population decline event did not have a significant effect on the genetic diversity of *T. mongolica*. The conservation status of *T. mongolica* is however, clearly under severe threat and this study indicates that urgent measures need to be put into place to ensure its ongoing survival.

### Population genetic structure

Landscape features such as rivers and mountains can function as geographical barriers to dispersal and gene flow, shaping population structure (*Funk et al., 2001*; *Whiteley, Spruell & Allendorf, 2004*). STRUCTURE analysis did not clearly identify genetic clusters corresponding to specific populations (Fig. 2). Clustering results indicated unobstructed admixture and thus weak genetic differentiation among *T. mongolica* populations. This result was corroborated by the pairwise $F_{ST}$ estimates and FCA analysis (Table 2, Fig. 3). Moreover, popgraph analysis showed that the genetic structure is weak, and all of the samples were not assigned to any genetic group, suggesting ongoing admixture processes between the extant populations (Fig. 4). For most angiosperms, nuclear genes are inherited paternally via pollen, and maternally via seeds, while cytoplasmic genes found in the chloroplast and mitochondria are maternally inherited (*Petit, Kremer & Wagner, 1993*). Complex configurations of gene flow within and among populations are expected through nuclear and chloroplast markers (*Petit et al., 2005*). In this study, the patterns of genetic structure inferred from nuclear microsatellite markers suggests that the Yellow River and Zhuozi Mountain do not act as significant barriers to pollination among populations.

The Yellow River, the second longest river in China, is well-known for its frequent flooding and heavy silt load (*Sinclair, 1987*). In the last 3,000 years, the river's levees

have breached more than 1,500 times and its course has changed approximately 26 times (*Leung, 1996*). As a result, *T. mongolica* populations on the flood plain have been exposed to periodic habitat destruction and fragmentation (*Ge et al., 2011*). It is established that species with narrow distributions and small population sizes face a high risk of extinction, especially when gene flow between sub-populations is restricted (*Frankham, Ballou & Briscoe, 2002*; *Hanski & Gilpin, 1997*). In seed plants, such gene flow occurs via the movement of pollen or seeds. Fortunately, *T. mongolica* is primarily pollinated by insects (*Zhen & Liu, 2008*), negating the potential barrier effect of the Yellow River and, to some degree, the Zhuozi Mountain (Fig. 1). Hence, neither distinguishable genetic clusters nor population differentiation were detected in populations separated by these barriers.

## Population demography

In present study, neither heterozygosity excess nor mode-shift tests suggested a recent population bottleneck for *T. mongolica*. However, MSVAR simulation indicated a severe recent population decline in all populations (Fig. 5). Under the exponential model, the posterior distribution of $N_0$ and $N_1$ (50% quantile) indicates a 67-fold population decline, starting approximately 5,455 years ago, and is mirrored in similar declines for animals. For example, in Northeastern Malaysia, human-induced deforestation and habitat fragmentation resulted in a recent population collapse in orangutans, *Pongo pygmaeus*, approximately 210 years ago (*Goossens et al., 2006*). Humans in southwestern China, over the course of thousands of years, have caused the dramatic decline of the giant panda, *Ailuropoda melanoleuca* (*Zhang et al., 2007*) and the tufted deer, *Elaphodus cephalophus* in the Yangtze River area (*Sun et al., 2016*). These events suggest the possibility of an anthropogenically-induced decline for *T. mongolica*. In addition, it is worth noting that the high-density human activities along the G6 road could also significantly impact the *T. mongolica* populations in the next few decades.

## Implications for conservation

Population genetics studies can help to identify management units (MUs) and evolutionarily significant units (ESUs) for conservation (*Moritz, 1994*). In this study, all of the analytical results indicate weak genetic differentiation among extant populations of *T. mongolica*. Our work suggests that the eight *T. mongolica* populations sampled may be deemed a single MU for conservation purposes. With rapidly increasing human disturbance, *T. mongolica* populations are suffering from overexploitation, habitat loss and fragmentation, most noticeably along the G6 road. To better maintain the population size of *T. mongolica*, we propose that the Chinese government should give greater priority to the conservation and restoration of its habitat, and to plant more artificial populations in the core area of its current range along the G6 Road.

## CONCLUSION

In this study, 339 individuals from eight populations were successfully genotyped at 12 nuclear loci, successfully. Based on microsatellite data, high levels of genetic diversity were revealed in this endangered species. This study implies that the wild *T. mongolica*

populations still harbor a surprisingly rich gene pool. Furthermore, neither distinguishable genetic clusters nor population differentiation were detected among extant *T. mongolica* populations. Finally, a strong and recent population decline event was discovered, which was likely to have been brought about by recent human activities, and emphasizes the need for urgent conservation measures to ensure its ongoing survival.

## ACKNOWLEDGEMENTS

The authors would like to thank Qiang Wang for his help in sample collecting. We would also like to thank Dr. Jessica Pérez-Alquicira (National Autonomous University of Mexico, Mexico), Dr. Tao Pan, Dr. Chaochao Hu, Li Qin and Ms. Guiyou Wu for their laboratory assistance and insightful advice. We would also like to thank Mrs. Lise K. Sorensen (University of Utah, USA) for her linguistic assistance.

### Funding

This study was supported by the Inner Mongolia Science and Technology Plan Foundation (20160415), Inner Mongolia Science and Technology Innovation Guide Fund (CX2016011 and CX2017008), Inner Mongolia Science and Technology Major Foundation (NK2016ZD1024), Inner Mongolia Industrial Innovation Talent Team (20130430, 20171903), Shenhua Shendong Group Science and Technology Foundation (CSIE-HT20162190), Ordos Science and Technology Foundation (20161220) and Ordos High-Level Talent Innovation and Venture Base Foundation (2016121901). The funders had no role in study design, data collection and analysis, decision to publish, or preparation of the manuscript.

### Grant Disclosures

The following grant information was disclosed by the authors:
Inner Mongolia Science and Technology Plan Foundation: 20160415.
Inner Mongolia Science and Technology Innovation Guide Fund: CX2016011, CX2017008.
Inner Mongolia Science and Technology Major Foundation: NK2016ZD1024.
Inner Mongolia Industrial Innovation Talent Team: 20130430, 20171903.
Shenhua Shendong Group Science and Technology Foundation: CSIE-HT20162190.
Ordos Science and Technology Foundation: 20161220.
Ordos High-Level Talent Innovation and Venture Base Foundation: 2016121901.

### Competing Interests

Yangnan Guo is employed by China Energy Technology Holdings Ltd.

### Author Contributions

- Yingbiao Zhi and Baowei Zhang conceived and designed the experiments, performed the experiments, analyzed the data, contributed reagents/materials/analysis tools, prepared figures and/or tables, authored or reviewed drafts of the paper, approved the final draft.

- Zhonglou Sun conceived and designed the experiments, performed the experiments, analyzed the data, prepared figures and/or tables, authored or reviewed drafts of the paper, approved the final draft.
- Ping Sun conceived and designed the experiments, performed the experiments, analyzed the data, prepared figures and/or tables.
- Kai Zhao conceived and designed the experiments, analyzed the data, prepared figures and/or tables, approved the final draft.
- Yangnan Guo conceived and designed the experiments, analyzed the data, prepared figures and/or tables, approved the final draft.
- Dejian Zhang prepared figures and/or tables, approved the final draft.

**Field Study Permissions**

The following information was supplied relating to field study approvals (i.e., approving body and any reference numbers):

Field experiments were approved by the West Ordos National Nature Reserve, Inner Mongolia Province, China.

**Data Availability**

The raw data are provided in a Supplemental File.

**Supplemental Information**

Supplemental information for this article can be found online at http://dx.doi.org/10.7717/peerj.5645#supplemental-information.

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
