# Peer review of "How much genetic variation is stored in the endangered and fragmented shrub Tetraena mongolica Maxim?"

_PeerJ, doi:10.7717/peerj.5645_

## Round 0.1 · original submission · Major Revisions

Dear Baowei,

Both reviewers agree on the significance of your study but also both raise many suggestions to improve the manuscript.

Reviewer #1 highlights a reference that should be taken into account. Both reviewers suggest that the writing could be significantly improved. And also suggest that methods should be better justified. Discussion could be also improved.

Please submit a new improved version following suggestions from both reviewers.

Cheers,

Marcial.

Reviewer 1 ·

Basic reporting

The manuscript is, in general, written in understandable English, but there are a number of grammatical errors which need to be made and improvements which I would like to suggest. I have indicated these errors and improvements in the general comments to Author.

The authors have used sufficient references, but they seem to have largely ignored the work of Ge et al., 2003. The aim of that study was the same as theirs, namely to assess the genetic variation in Tetraena mongolica, although different methods were used for that assessment, i.e. ISSRs. Ge et al., also come to very similar conclusions, and these are also not stated in the Introduction, nor are they discussed in the Discussion. The Conclusions that Ge et al., 2003 are furthermore also very similar, both with regard to a recent bottleneck in the population and that the populations are highly endangered. If this was the recommendation already in 2003, can the authors comment whether anything has changed? Certainly, the work done by Ge et al., 2003, must be discussed in the present publication both in the context of the similarity of their findings in spite of the use of different methods, and the consequences of their findings and whether the previous recommendations have at all been implemented, which appear not to be the case.

The article conforms to the professional article structure including figures and tables. Raw data has not been shared. Aims and objectives have been clearly stated.

Experimental design

The research described in this manuscript does conform to PeerJ's requirement of scientific and methodological soundness.

In the light of the comments made about the fact that this study has the same aims as the study of Ge et al., 2003, it is somewhat questionable whether the study is meaningful and relevant, and because the findings do not contradict the findings of Ge et al., 2003, nor do they really fill a knowledge gap, all they do is they reiterate and confirm the study of Ge et al., 2003, albeit with different experimental methods and therefore different results.

The experimental design in this study is relatively simple, i.e. sampling of extant populations and microsatellite analysis followed by the population genetic analysis based on the generated data. The methods are described in sufficient detail to allow their replication. The microsatellite analyses appear to have been correctly performed and the results have been correctly presented and interpreted.

Validity of the findings

The microsatellite results of the study are new, and the data appears to be robust and statistically sound. The results represent a replication of a previous study which analysed the same research question using different experimental methods. The conclusions are well stated and clear. The authors have not speculated beyond the limits of their data. This study is, however, not adequately discussed in the context of the previous study by Ge et al., 2003. Both studies come to the same basic conclusions.

Additional comments

The authors state that the genus is monotypic (line 62) which is incorrect . Beier et al., 2003 performed a phylogenetic analysis on a large proportion of the members of the subfamily Zygophylloideae and renamed approximately 40 species, previously placed in the genus Zygophyllum, in a broader genus Tetraena. This is the currently accepted taxonomy of the genus. The authors should correct this and reference relevant recent literature confirming this (Lauterbach et al., 2016).

On line 68, the authors state that the shrub contains triacylglycerol. Firstly, triacylglycerol should be replaced with "triacylglycerols" i.e. plural. Triacylglyerols are the chemical name for fats consisting of glycerol esterified with fatty acids of different lengths and different numbers of double bonds i.e. saturation. They are always a mixture and should be referred to in plural. A reference should be given. If this is an assumption, then I would suggest that "triacylglycerol" be replaced with "waxes and oils", as these are the most common highly combustible compounds that are found in plants.

In line 143, I would like to question the validity of the average generation time really being as low as only 4 years. These are perennial woody shrubs with long lifespans. Unless they are harvested by humans as a result of which their generation time is lowered, I would question this assumption. How did they authors who have published this, come to this conclusion? In my opinion, this should at least be critically discussed in a little more detail.

Grammatical corrections and suggested improvements:
1. Title add "the" into the title: How much genetic variation is stored in the endangered and fragmented shrub Tetraena mongolica Maxim?
2. Line 54, replace "Besides" with "However".
3. Line 58, replace "Besides" with "Furthermore".
4. Line 96, add words: "purified by using the EasyPure etc."
5. Line 98: "genetic markers".
6. Line 99: I question whether MgSO4 was used, was MgCl2 not rather used?
7. Line 104: "with the GS500 marker".
8. Line 113: replace "Besides" with "In addition".
9. Line 168: "psendophase" should be "pseudophase".
10. Line 177: replace "meanwhile" with "whilst" and remove comma before and after the word.
11. Line 179: replace "disclosed" with "indicated".
12. Line 180: insert "a" to read "passed through a significant reduction".
13. Line 182: insert "the current" in "for the current T. mongolica population".
14. Line 184: replace "times" with "fold".
15. Line 186: replace "indicating" with "indicate". Also replace "in" with "approximately".
16. Line 189: insert "a" in "has undergone a dramatic population".
17. Line 238: correct and expand "siginificant" should read "significantly".
18. Line 240: correct "genetics" to "genetic".
19. Line 242: I would suggest to add "in T. mongolica" after "among extant populations".
20. Line 250: correct sentence to read "were successfully genotyped at 12 nuclear loci".
21. Line 255: correct sentence "which was likely to be affected by early mankind activities" to read "which was likely to be affected by recent human activities".
22. Line 313: the author name should be written in capitals "Maxim".
23. Line 334: capitalize journal name.
24. Line 348: do not abbreviate journal name.
25. Line 353: add "of the USA" to journal name.
26. Line 361: Write species name correctly "Adiantum capillus-veneris".
27. Line 389: correct "Republicae".
28. Figure 1 legend "populations".
29. Table 2 Legends: "shrubs"

·

Basic reporting

The grammar should be checked, as there are some very long running sentences. Not all the references in the text correspond to the appropriate figure or table. Please, make sure all references are correct.

Experimental design

1.- Field sampling and work were approved
2.- It was mentioned that the species occupies a wide area in the Gobi desert, but is it abundant? Are those 8 populations representative of the entire range? What is the entire range?
3.- The estimation of null alleles has to be mentioned in the methods
4.- l. 126 The priors in the Structure analysis are correct, but more explanation on why they were chosen would be useful.
5.- In general, please state clearly how the significance of each result was measured and that all appropriate p-values are clearly reported.

Validity of the findings

1.- Table 3.- There are no double asterisks, please correct the legend on the table. Besides, if the comparisons are all significantly different from 0 in all pairs with asterisks, then the authors cannot argue extremely weak genetic differentiation (l. 171), as it is low, but significant in most cases. The authors mention there is no significant IBD, but refer to Fig.3, where there is no information on the tests. Please include the IBD graph, correlation coefficient and p-values for that test in the results section.
2.- There are some contradicting statements in the discussion regarding to the Structure and Fst results. Whereas there is no broad geographic pattern of population structure, the results of both analyses show that populations are differentiated due to differences in allele frequencies. A visual inspection of the raw data supports this idea, therefore one cannot just infer widespread genetic connectivity and gene flow from the results. Structure can detect very broad patterns of genetic structure, and FST is a summary statistic with limited explanatory power. Therefore, I suggest to do a multivariate analyses based on genetic covariance like the one implemented by Population Graphs (Dyer & Nason 2004). This type of graph would show which populations are indeed genetically connected to one another, and which could be vulnerable to isolation (limited number of connections).( http://dyerlab.github.io/popgraph/).
3.- As one of the aims of this work was to evaluate the effect of landscape or antropogenic barriers in the distribution of genetic variation among populations, I consider that the structure analysis without any geographic information may not be the most accurate. I suggest an analysis designed to detect genetic barriers, such as Monmonier implemented by the R-package adegenet (Jombart 2008). The results on their own do not really put to test the main premise of the effects of barriers on the genetic variation of the populations, and the significant pairwise Fst values suggest there might be genetic differentiation among different populations that is not clear in the STRUCTURE analysis.

Additional comments

The work has been through and the sampling quite impressive. I believe these types of studies are very important to evaluate the status in rapidly declining populations. However, the the analyses implemented were not the appropriate ones to detect what might be weak signals of genetic differentiation among populations.

---

## Round 0.2 · Minor Revisions

Dear Baowei,

There is a significant improvement of your manuscript but still one of the reviewers has concerns about your study.

Please, submit a new version following the suggestions by reviewer 1.

Sincerely,

Marcial.

Reviewer 1 ·

Basic reporting

After the first review and the implementation of suggested changes, as well as the linguistic assistance given to improve the manuscript, it is now a lot clearer. The authors have expanded the literature references which is good. All figures and tables are correctly prepared and presented.

Experimental design

After review, the experimental design now conforms to the requirements of PeerJ.

Validity of the findings

The findings are valid, but I have my concerns that the SSR technique gives an overestimation of genetic diversity. I have therefore suggested certain additional changes to the interpretation of these data. I am also concerned that the conservation message of the findings of this study are underestimated, and the study sketches a very poor picture for the long-term survival of the species. I have therefore suggested some modifications which emphasize this more strongly. My modifications and corrections are contained in the attached file.

Additional comments

I am satisfied that the changes that I have requested have been made by the authors. However, through additional changes that the authors have made, further errors and misinterpretations have now crept in again which require corrections once again. I have made suggestions for improvements and once these have been made, the manuscript will be in a publishable form.

Annotated reviews are not available for download in order to protect the identity of reviewers who chose to remain anonymous.

·

Basic reporting

The grammar was greatly improved.

Experimental design

No comment

Validity of the findings

The new analyses supported previous conclusions. There is no more to add.

Additional comments

The authors have greatly improved the manuscript and addressed all previous concerns.

---

## Round 0.3 · accepted · Accept

Dear Baowei,

Congratulations! Your manuscript has been accepted for publication in PeerJ.

Cheers,

Marcial Escudero.

#